# Trust in the Work Environment and Cardiovascular Disease Risk: Findings from the Gallup-Sharecare Well-Being Index

**DOI:** 10.3390/ijerph16020230

**Published:** 2019-01-15

**Authors:** Toni Alterman, Rebecca Tsai, Jun Ju, Kevin M. Kelly

**Affiliations:** 1Division of Surveillance, Hazard Evaluations and Field Studies, National Institute for Occupational Safety and Health, CDC, (MS-R17), 1090 Tusculum Ave, Cincinnati, OH 45226, USA; vht5@cdc.gov (R.T.); jnj7@cdc.gov (J.J.); 2UI Healthier Workforce Center, The University of Iowa, UI Research Park, IREH #106, Iowa City, IA 52242, USA; kevin-kelly@uiowa.edu

**Keywords:** cardiovascular disease, work environment, social capital, trust, Total Worker Health^®^, health behaviors, job stress

## Abstract

This study examined associations between trust, an important aspect of workplace social capital, with seven cardiovascular disease (CVD) risk factors (American Heart Association Life’s Simple 7 (LS7)): smoking, obesity, low physical activity, poor diet, diabetes, high cholesterol, and high blood pressure. Data are from the U.S. Gallup-Sharecare Well-Being Index (2010–2012), a nationally representative telephone survey of U.S. workers (*n* = 412,884). The independent variable was the response to a work environment (WE) question as to whether their supervisor always creates an open and trusting environment. Regression models were adjusted for demographic characteristics with each of the LS7 CVD risk factors as dependent variables. Twenty-one percent of workers reported that their supervisor did not create an open and trusting environment. Trust was associated with increased adjusted odds of having many of the LS7 CVD risk factors. Among those workers whose supervisor created a mistrustful environment, the odds ratios were greatest (>20%) for having four or more of the LS7 CVD risk factors.

## 1. Introduction

Cardiovascular disease (CVD) continues to be a costly and significant problem and is the leading cause of death in the United States [1]. Moreover, by 2030, the prevalence of CVD among American adults (20 years of age and older) is expected to increase from 35% to over 40% [2]; direct medical costs of CVD are expected to triple to $818 billion [2,3]. To address these pressing issues, the American Heart Association (AHA) set strategic impact goals to improve cardiovascular health by 20% and achieve a 20% reduction in CVD mortality by 2020 [4,5,6].

Numerous studies have examined associations between work stress and CVD [7,8,9,10,11,12,13,14,15,16]. In addition, many studies have examined associations between work organization and workplace psychosocial factors with CVD and its risk factors [17,18,19,20,21,22,23,24]. However, much of the occupational health literature on CVD has focused on a few select models such as job demand and control [11,19,20]; job demands-resources [25,26,27]; social support [18]; and effort–reward imbalance [17,23]. In addition, attention has more recently been given to the role of work engagement and cardiovascular reactivity [28] and forms of organizational justice that share some aspects of the effort-reward model [29,30,31,32,33].

Cardiovascular health can be assessed by AHA’s My Life Check^®^ Life’s Simple 7 (LS7) [4,5,6]. The AHA defined ideal cardiovascular health by the presence of all four favorable health behaviors (abstinence from smoking within the past year, ideal body mass index (BMI), physically active, and healthy diet) and three favorable health factors (ideal fasting glucose, ideal total cholesterol, and ideal blood pressure). Having ideal levels in all seven components of LS7 can increase life span and reduce healthcare costs [4,5]. 

Recent literature has focused on the theory of social capital as important in explaining these health behaviors. While there are many definitions of social capital [34,35,36,37,38], this study uses a relational or social cohesion approach suggested by Berkman and Kawachi [39], who define social capital ‘as those features of social structures such as levels of interpersonal trust and norms of reciprocity and mutual aid—which act as resources for individuals and facilitate collective actions’. Measures of social capital involve examining elements of a relationship, relational networks, levels of trust, and levels of collaborative activity [40].

In the past, literature on associations between social capital and health focused mainly on community, residential, or geographic areas [36,41,42,43]. More recently, workplaces have been seen as important social units where social capital may promote well-being and health and as providing a means of understanding relationships in the workplace [44,45,46,47,48]. A number of hypotheses as to how social capital may act on health behaviors have been proposed; these include providing norms and attitudes that influence health behaviors, and psychosocial mechanisms that promote emotional support and enhance self-esteem [39]. For example, Lindström and Giordano [49] suggest that social capital reduces cigarette smoking by (1) deterring socially ‘deviant’ behavior; (2) increasing dissemination of positive health messages; and (3) providing a buffer against psychosocial stress’. 

Some findings of associations between social capital and health behaviors have been mixed [34,50]. A recent systematic review of 14 prospective studies using a variety of definitions of social capital and different contexts found no association among most social capital dimensions and all-cause mortality, CVD, or cancer [34]. However, definitions of social capital varied among the individual studies reviewed, including dimensions of social participation, social network, civic participation, social support, trust, norms of reciprocity, and sense of community [34]. Other empirical research supports associations between social capital and health, including mental health [42,51,52]; diet [53]; alcohol use [54,55,56]; physical activity [57,58,59]; hypertension [60]; and smoking [49,61,62,63]. A recent study by Nieminen et al. [64] found support for an association between social capital and five health behaviors (smoking, alcohol use, physical activity, vegetable consumption, and sleep). Analyses of data from the Finnish Public Sector Study found that low workplace social capital was associated with the co-occurrence of multiple lifestyle risk factors in cross-sectional analyses, but not in longitudinal analyses [65]. 

The report of the 2017 Total Worker Health^®^ Workshop [66] identified “perceived working conditions and supervisor support,” the bases of worker trust, as important worker-level measures for understanding worker health. Trust is acknowledged as a key principle in the supervisor–subordinate relationship, especially as it as it relates to the distribution of rewards, sanctions, and resources [67] including promotions, pay raises, and job security [68]. Moreover, Schill [69] reminds us that “leaders at all levels set the tone (for Total Worker Health) through their shared commitment to safety, health and well-being”. There are many definitions of trust, but for the current study, the authors define trust as a multidimensional psychological state that involves cognitive processes as well as affective and motivational components [68]. For trust to develop, there needs to be understanding, fairness, and mutual respect between the supervisor and subordinate. 

It is often difficult and expensive to collect data on work environment and workplace psychosocial factors across multiple worksites and regions. In the U.S., a number of ongoing national surveys, such as the Quality of Worklife [70], the Health and Retirement Survey (HRS) [71], and the 2010 [72] and 2015 National Health Interview Survey (NHIS) [73], have included work organization and workplace psychosocial questions. The Gallup-Sharecare Well-Being Index (WBI) [74] collects data from adults 18 years and older living in the United States, including questions on work environment (WE). A number of studies have used the Gallup survey to look at health and well-being [74,75,76,77,78], but few have specifically focused on the work environment questions associated with social capital in relation to health.

The current study examines whether trust, an important aspects of social capital, is associated with the seven CVD risk factors identified in the AHA LS7 screening tool. Increasing social capital may improve health behaviors and outcomes directly, or in conjunction with workplace prevention and intervention programs. Due to the gender differences in the prevalence, progression, and underlying mechanisms in CVD, results are presented separately for women and men [79].

## 2. Materials and Methods

### 2.1. Data Source

Data for this study are based on the Gallup-Sharecare Well-Being Index (WBI) survey conducted between 2010 and 2012 in the United States. Every day, the Gallup Organization conducts computer-assisted telephone interviews (in English or Spanish) with 1000 randomly sampled U.S. adults (≥18 years of age) on political, economic, and well-being topics. Random-digit-dial (RDD) to landlines and cell phones was used to reach wireless-only and wireless-mostly households. Although the response rate is low, 9–11%, it is estimated that the Gallup sample covers more than 95% of the U.S. adult population. Gallup weights data daily to account for disproportionate selection in age, sex, geographic region, gender, education level, ethnicity, race, self-reported location, and phone use status [73].

### 2.2. Study Population

The sample consists of survey participants interviewed between 2010 and 2012 who reported being currently employed by an employer for at least 30 h per week. Only workers who were employed by an employer were selected because this study focuses on supervisor behavior. In addition, full-time workers (i.e., those working 30 or more hours per week) were selected because results from this study would be more relevant to those who spend a greater proportion of their time at work, than those working part time.

### 2.3. Measures

Life’s Simple 7: The WBI included questions that address the seven cardiovascular health components of the LS7, but are not a complete match to the AHA definition. Figure 1 shows how the authors adapted WBI questions to match AHA’s LS7 components. For blood pressure, the WBI asks if the respondent had ever been told by a physician or nurse that they had high blood pressure (yes = high blood pressure). The LS7 defines high blood pressure as >120/80 mm Hg. Similarly, the WBI asks if the respondent had ever been told by a physician or nurse that they had high cholesterol, (yes = high cholesterol). The LS7 defines high cholesterol as total cholesterol >200 mg/dL. Diabetes in the WBI is measured by the response to a question about whether the respondent had been told by a physician or nurse that they had diabetes (yes = diabetes). The LS7 defines diabetes as a fasting plasma glucose >100 mg/dL. The study definition for at-risk BMI (based on self-reported height and weight), or obesity, was BMI ≥ 30 kg/m^2^; the LS7 definition for at-risk BMI is ≥25 kg/m^2^. For diet, the WBI asks respondents the number of days per week they consumed five or more servings of fruits or vegetables. For this study, a healthy diet was defined as consuming five or more servings of fruits or vegetables seven days per week; adequate consumption on less than seven days was defined as a poor diet. The LS7′s dietary assessment requires additional information such as the amount of fish, sodium, whole grains, and sugary beverages consumed daily or weekly. Similarly, for physical activity, the WBI asks about the number of days of exercise for at least 30 min during the past week, but does not ask about the intensity of physical activity. The LS7 definition includes the amount of vigorous and moderate activities per week. In the current study, meeting physical activity guidelines was defined as exercising for at least 30 min five days per week. Performing less than this was defined as insufficient exercise. In the WBI, smoking is defined as a yes response to a question about whether the respondent smokes. In the LS7, smoking is based on whether the person is a current smoker or quit <12 months ago.

Work environment: Gallup scientists created a work environment question related to social capital based on their understanding of the literature. Work environment was coded in a negative direction and measured by the question: ‘Does your supervisor always create an environment that is trusting and open, or not?’ (No = mistrustful environment).

### 2.4. Data Analysis

All analyses were stratified by gender and calculated using SAS 9.3 (SAS Institute Inc., Cary, NC, USA), using weights provided by Gallup to represent the U.S. population and to account for the complex survey design. Gallup surveyed 1,059,894 individuals between 2010 and 2012, of which 412,884 were full-time employees (working 30 or more hours per week). Responses of ‘don’t know’ or ‘refused’ were set to missing. Because of a skip pattern, 23,554 (6%) were not asked the open and trusting environment question. Approximately 14% of the sample were missing income data, and 5% or fewer were missing data for the LS7 risk factors. Descriptive statistics were calculated for demographic, LS7, and the trust question. *Z*-tests for the difference between women and men were calculated. Logistic regression models were run with each of the LS7 CVD risk factors as dependent variables in separate regression models. Odds ratios and 95% confidence intervals were calculated. Confidence intervals excluding 1.0 indicate significance at *p* < 0.05. In addition, a sensitivity analysis was conducted to examine the impact on associations using four or more CVD risk factors as a dependent variable in the regression model. Trust was entered into separate logistic regression models as an independent variable. All models were adjusted for potential confounders including demographic factors: age (years); race/ethnicity (White, Black, Asian, Hispanic, and Other); education (less than high school diploma, high school graduate, technical/some college or associate degree, college degree, and post-graduate degree); marital status (single/never married, married, separated, divorced, widowed, and domestic partner); family income (<$1000/month, $1000–$2999/month, $3000–$4999/month, $5000–$7499/month, and ≥$7500/month) and any health insurance. Due to the large sample size in the current study, we focused on effect sizes (odds ratios) rather than *p*-values. According to a recent policy statement by the American Statistical Association on *p*-values and statistical significance, any effect, no matter how tiny, can produce a small *p*-value if the sample size is large enough [80].

## 3. Results

Demographic characteristics: Forty-two percent of the sample were women. Descriptive statistics for workers stratified by gender and trust are presented in Table 1. For both women and men, the largest percent of respondents (72.0% and 72.2%, respectively) were White, Non-Hispanic with a mean age of about 42. More than half of the respondents were married and had either technical training or some college.

Mistrustful environment: Approximately 22% of women and 20.3% of men indicated that their supervisor did not always create an open and trusting environment (Table 1). As shown in Table 1, for both women and men, the highest prevalence of mistrust reported was among workers ages 45–64, (women = 24.4%, men = 23.0%) followed by workers ages 30–44 (women = 22.3%, men = 20.5%). Black women (23.2%), followed by White women (22.4%) had the highest prevalence of reporting that their supervisor does not create an open and trusting environment, compared with women of other races/ethnicities (Asian, Hispanic, Other). White men (21.1%) followed by Black men (20.6%) reported higher prevalence of a mistrustful environment. Prevalence of a mistrustful environment was higher for women with increasing levels of education (highest for those with college or post-graduate education, 25.2%). Men with technical training or some college/associate degree had a slightly higher prevalence (20.9%) of a mistrustful environment. Divorced women (26.8%) and men (24.7%) had the highest prevalence of reporting a mistrustful work environment. Prevalence by income was similar for both women and men, with slightly higher prevalence of a mistrustful environment for those earning $3000–$4999 per month among women (23.0%) and for those earning $5000–$7499 among men (21.7%).

Table 2 shows the weighted prevalence of LS7 risk factors by trust. The prevalence of all LS7 risk factors were higher for both women and men who reported working in a mistrustful environment, compared to those whose working environment was not mistrustful.

Multivariate results for trust with LS7 CVD risk factors, after adjustment for demographic factors and having any health insurance, are shown in Table 3. Confidence intervals excluded 1.0, indicating statistical significance at *p* < 0.05 for each outcome. Trust was associated with the LS7 CVD risk factors in both men and women after adjustment for covariates. Due to the large sample size, we report effect sizes with a focus on those ≥10%. We found that workers who did not work in an open and trusting environment had greater odds of having high blood pressure (women = 15%, men = 20%), high cholesterol (women = 18%, men = 22%), and diabetes (women = 15%, men = 18%) compared to those who reported having an open and trusting environment with their supervisor. Both women and men workers had greater odds of being a current smoker (both 15%), having a poor diet (women = 10%, men = 11%), and being obese (both women and men = 18%). Women reporting a mistrustful environment also had greater odds of having a low physical activity level (10%). Odds ratios for having four or more LS7 CVD risk factors were elevated for those working in a mistrustful environment (women = 22%, men = 29%).

## 4. Discussion

The findings of this study suggest that lower workplace social capital, as measured by the WBI, is associated with higher odds of having one or more of the LS7 CVD risk factors.

Our findings are consistent with others who have found associations between social capital and health [39,46,48,65,81,82,83]. Previous research reported that working in a negative environment and having low social support could lead to stress and psychosocial distress [84]. Workplace stress can directly increase CVD risk through biological pathways (e.g., inflammation) or CVD risk factors [12,13]. Studies have found that workers experiencing job stress were more likely to have diabetes [85]. Associations between work environment and high blood pressure and high blood cholesterol are mixed, with some studies reporting results similar to our findings [86,87] and others reporting no association [85].

Furthermore, workplace stress can indirectly affect CVD risk through at-risk health behaviors [85,87]. These behaviors include poor diet, insufficient physical activity [85,86], smoking [85], high alcohol consumption, and lack of sleep [88]. Workers who reported a lack of support from supervisors were more likely to be heavy smokers [89]. In addition, men with low workplace social support were more likely to be obese [90].

Although we adjusted for potential demographic confounders in our models, we examined each of the LS7 CVD risk factors separately. Health behaviors are often interrelated and can affect the presence or absence of other health behaviors. For example, smoking was found to increase caloric intake [91], while a healthy lifestyle (diet and exercise) was negatively associated with smoking [92]. Because of the potential co-occurrence, we conducted sensitivity analyses to see whether the odds ratios increased if we selected having four or more of the LS7 risk factors instead of only one. We used this as the dependent variable in our regression models. Both women (22%) and men (29%) showed an increase in odds for having four or more LS7 risk factors if they indicated that their supervisor did not create an open and trusting environment.

Analyses were presented separately by gender, not only due to the differences in CVD risk between women and men, but also due to the importance of gender in the social capital and managerial psychology literature [93]. Odds ratios were similar for both genders when the LS7 factors were looked at individually, and slightly higher among men when the dependent variable was having four or more LS7 factors.

Improvements to the work environment are needed to reduce CVD risk among workers. Social modification to the work environment, such as adjusting managerial style to create an open and trusting environment, can decrease work stress. Considering managerial trust from a Total Worker Health^®^ framework meets the goals of illness prevention to advance worker well-being. Efforts can also be made to target the health behaviors themselves. There are a range of possible strategies for addressing the LS7 risk factors in the workplace. For example, physical modification to the work environment, such as installing sit/stand desk stations and walking workstations, can reduce sedentary behavior and may increase physical activity. Additionally, increased access to nutritious food in the workplace may improve diet. Supervisors who support workplace wellness may help in reducing CVD risk factor in workers [94].

### Strengths and Limitations

This study has several strengths. The WBI is a large, nationally representative survey. Skopec et al. [95] found that the survey provided reasonably similar data when compared to established national surveys, such as the National Health Interview Survey (NHIS) and the Behavioral Risk Factor Surveillance System (BFRSS), on several important health-related measures. However, the Gallup sample was slightly older, had fewer minorities, and a higher educated sample than in other national surveys [95]. Outcomes examined included select health behaviors and health outcomes. Findings in our study are similar to those reported by adults who worked in the past 12 months in the 2010 National Health Interview Survey (NHIS) conducted by the National Center for Health Statistics. Weighted prevalence items were obesity (BMI ≥30) (Gallup = 27.7%; NHIS = 28.1%), current smoker (Gallup = 21.9%, NHIS = 19.7%), hypertension (Gallup = 22.3%; NHIS = 19.4%), insufficient exercise (Gallup = 76.3%; NHIS = 88.3%), and diabetes (Gallup = 6.1%; NHIS = 5.8%) [72]. Data on high cholesterol are not available for the 2010 NHIS, but are available for 2015 (Gallup = 21.3%; NHIS = 21.5%) [72].

The work environment question included in this study allows us to examine an important workplace psychosocial factor that is often difficult or expensive to study. It is unclear where the work environment question originated. Documentation provided by Gallup indicated that it was based upon findings from leading scientists in the areas of survey research, behavioral economics, and health. No information on the validity or reliability of the Gallup question is available.

This study also has several limitations. The survey is cross-sectional and therefore no conclusions can be made regarding causality. Data were collected via a telephone survey that has a low response rate, potentially affecting the representativeness of our findings. For each regression model, observations with missing values for included covariates were dropped. All data were self-reported at one point in time and are subject to response biases, such as recall and social desirability. Social desirability bias [96] is the tendency of respondents to present themselves in a socially desirable light, which may deviate from their true behaviors. Social desirability bias has been shown to affect the reporting of health behaviors, including underreporting negative behaviors and over reporting positive ones [97]. However, a recent study by Prather et al. [98] did not find confounding due to social desirability bias. Additionally, although we adjusted for potential confounders in our models, other non-measured confounders may have influenced our results. The WBI survey only touched upon a small number of components of an individual’s work environment. Components of social capital and the work environment such as occupation, organizational structure (e.g., work schedule, work arrangements), culture, job autonomy, job resources, job security, work engagement, workplace hostility, additional characteristics of the supervisor, and others are needed for a better-informed study. Findings by Oksanen et al. [45,46] suggest that the effects of low social capital might not be similar in all work units or groups of different socioeconomic structure. However, because social capital and socioeconomic status were measured at the individual level, we are unable to examine the effects of social capital in different work contexts. Additionally, stand-alone, single-item questions may not offer the precision needed to make an accurate assessment of supervisory style, and as Choi et al. [34] suggest, there is a lack of consensus on measurement of social capital. The measure of social capital used in this study deals with leadership trust. Researchers have also included differing measures of social capital that include employee networks and workforce norms [99].

Health behavior questions in the WBI were different from AHA’s LS7 definitions, particularly diet that included only one of the five diet variables. The health factor metrics are markedly different in the WBI compared to the AHA’s LS7 definitions, noticeably absent are the clinical measurements of blood pressure, blood cholesterol, fasting glucose, and medication used to treat these health factors. Lastly, the study’s large sample size increases the probability of finding statistically significant associations; therefore, we focused on effect size rather than *p*-values. Despite these limitations, results show that more than 20% of workers report that their supervisor does not always create an open and trusting environment. This is associated with a 20% increase in odds for having four or more CVD risk factors, suggesting that this is an important factor when designing interventions to address worker cardiovascular health. Therefore, these results show support for the usefulness of this aspect of social capital to understand the work environment, supervisory behavior, and their association with worker cardiovascular health.

## 5. Conclusions

This study found that a negative work-environment characteristic representing an aspect of workplace social capital contributed to greater odds of having important CVD risk factors among full-time workers. Results suggest that supervisor behavior can play an important role in improving worker health. Workplace intervention programs for CVD and other chronic health conditions should consider addressing this aspect of workplace social capital, and supervisor competencies and behavior in particular, with proper training as a potential means to improve worker health. Thus, our results reinforce the notion voiced elsewhere [69,100] that supervisor support is essential to a comprehensive approach to worker safety and health; issues of managerial trust are worthy of inclusion in a Total Worker Health^®^ framework.

## Figures and Tables

**Figure 1 ijerph-16-00230-f001:**
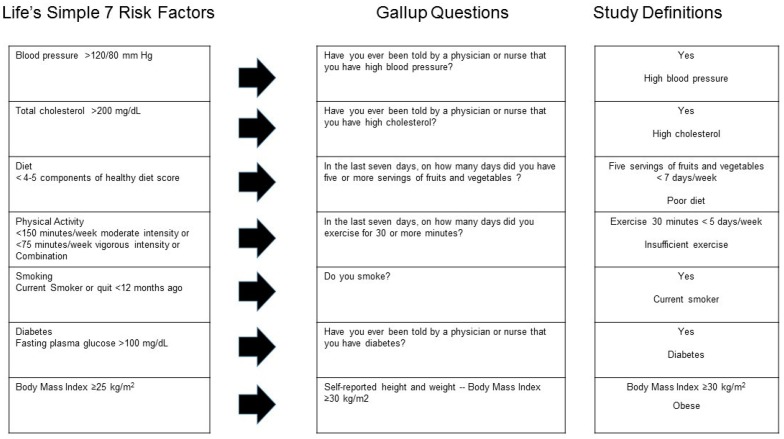
Cross-walk between American Heart Association Life’s Simple 7 (LS7) cardiovascular disease (CVD) risk factors and Gallup-Sharecare Well-Being Index.

**Table 1 ijerph-16-00230-t001:** Weighted prevalence (%) and standard errors (SE) of open and trusting and open work environment factors by sociodemographic characteristics and gender (Gallup-Sharecare Well-Being Index, 2010–2012).

Characteristics	Total Women	Mistrustful Environment		Total Men	Mistrustful Environment	
	%	%	SE	%	%	SE
Total		22.1	0.1		20.3	0.1
Age						
18–29 *	19.8	17.4	0.3	21.8	15.7	0.2
30–44 *	32.4	22.3	0.2	35.1	20.5	0.2
45–64 *	45.0	24.4	0.2	40.3	23.0	0.2
65+	2.8	17.2	0.5	2.9	15.9	0.5
Race/ethnicity						
White *	72.0	22.4	0.1	72.2	21.1	0.1
Black *	12.9	23.2	0.4	8.8	20.6	0.4
Asian	2.4	17.7	0.8	2.7	17.1	0.6
Hispanic *	9.9	18.7	0.4	12.8	16.3	0.3
Other	2.8	21.9	0.8	3.4	19.5	0.6
Education						
<High school *	4.4	19.3	0.7	7.1	17.3	0.5
High school graduate	21.9	20.0	0.3	26.0	19.8	0.2
Technical/some college or associate degree *	53.5	22.0	0.2	53.4	20.9	0.1
4-year college/post graduate *	20.3	25.2	0.3	16.6	20.4	0.2
Marital status						
Single/ Never married *	22.7	20.8	0.3	23.1	18.0	0.2
Married *	54.2	21.4	0.2	61.8	20.7	0.1
Separated	2.6	23.3	0.8	1.9	21.9	0.9
Divorced	12.1	26.8	0.4	7.0	24.7	0.4
Widowed *	2.9	22.8	0.7	0.8	20.8	1.1
Domestic partner *	5.5	23.2	0.6	5.5	20.1	0.5
Family Income per month						
<$1000 *	3.3	21.0	0.8	2.8	18.6	0.8
$1000–$,2999 *	27.0	22.5	0.3	22.9	20.1	0.3
$3000–$4999 *	26.6	23.0	0.3	23.8	21.2	0.2
$5000–$7499 *	21.4	22.5	0.3	21.5	21.7	0.2
≥$7500 *	21.9	21.9	0.3	29.1	19.6	0.2

* *Z*-test for difference between women and men *p* < 0.05.

**Table 2 ijerph-16-00230-t002:** Weighted prevalence (%) and standard errors (SE) for LS7 CVD risk factors by work environment characteristics and gender (Gallup-Sharecare Well-Being Index, 2010–2012).

**Mistrustful Environment**	**High Blood Pressure ***				**High Cholesterol ***				**Diabetes**				**Current Smoker ***			
	**Women**		**Men**		**Women**		**Men**		**Women**		**Men**		**Women**		**Men**	
	**%**	**SE**	**%**	**SE**	**%**	**SE**	**%**	**SE**	**%**	**SE**	**%**	**SE**	**%**	**SE**	**%**	**SE**
Yes	21.9	0.3	25.7	0.3	19.7	0.2	24.6	0.2	6.9	0.2	7.0	0.1	18.7	0.3	23.5	0.3
No	18.7	0.1	21.2	0.1	16.6	0.1	20.2	0.1	5.9	0.1	5.8	0.1	17.2	0.1	21.6	0.1
**Mistrustful Environment**	**Poor Diet ***				**Insufficient Exercise ***				**BMI ≥ 30 ***							
	**Women**		**Men**		**Women**		**Men**		**Women**		**Men**					
	**%**	**SE**	**%**	**SE**	**%**	**SE**	**%**	**SE**	**%**	**SE**	**%**	**SE**				
Yes	69.3	0.3	76.6	0.2	77.7	0.3	73.2	0.3	26.2	0.3	31.0	0.3				
No	67.6	0.2	75.1	0.1	76.1	0.1	71.4	0.1	22.8	0.2	26.9	0.1				

* *Z*-test for difference between women and men significant *p* < 0.05.

**Table 3 ijerph-16-00230-t003:** Multivariate associations between LS7 CVD risk factors and open and trusting work environment stratified by gender (Gallup-Sharecare Well-Being Index, 2010–2012) ^a^.

CVD Risk Factors (Dependent variables)	Models	
	Mistrustful Environment	
	OR (95% CI)	
	Women	Men
High blood pressure	1.15 (1.11, 1.20)	1.20 (1.16, 1.24)
High cholesterol	1.18 (1.13, 1.22)	1.22 (1.18,1.26)
Diabetes	1.15 (1.09, 1.23)	1.18 (1.12, 1.25)
Current smoker	1.15 (1.10, 1.20)	1.15 (1.11, 1.19)
Poor diet	1.10 (1.07, 1.14)	1.11 (1.07, 1.15)
Insufficient physical activity	1.10 (1.06, 1.14)	1.08 (1.05, 1.11)
Obese	1.18 (1.14, 1.23)	1.18 (1.15, 1.22)
Four or more risk factors	1.22 (1.16, 1.27)	1.29 (1.25, 1.34)

^a^ Models are adjusted for age, race/ethnicity, education, marital status, family income, and any health insurance.

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
