# Peer review of "Trust in the Work Environment and Cardiovascular Disease Risk: Findings from the Gallup-Sharecare Well-Being Index"

_ijerph, 2019, doi:10.3390/ijerph16020230_

Reviewer 1 Report

Dear Authors

Thank you for an interesting paper.

My main concern is that the title is very confusing and I spent a while trying to determine what the paper is actually about and was only clear when I saw the tables.

I suggest that you change the title to Trust in the work environment and cardiovascular disease risk factors.

As you indicate in the introduction and background that the conceptualisation of social capital is differently defined by different authors, depending on their conceptual framework. You also use the terms trust, social capital, social connectiveness, role of work environment ... I would stay clear of all this by not calling your paper Trust Social Capital...and not referring to your findings as social capital.

Figure 1 is missing.

Data

I would prefer to see some testing of significant differences between makes and females in Table 1 and Table 2.

Please provide more data in your report of the regression with indications of the Beta, significance and ORs.

Author Response

Thank you for your comments. We have changed the title to “Trust in the work environment and cardiovascular disease risk factors” as you suggested. We have also added Figure 1. In addition, we calculated Z-tests for the difference between women and men in Tables 1 and 2, and added asterisks and a footnote to indicate statistical significance at p<0.05. The text referring to Table 3 is now located near the table. We indicate in both the methods and results that 95% confidence intervals that exclude 1.0 are statistically significant at p<0.05. Although it is typical in linear regression to provide the beta values, in multivariable logistic regression, because the odds ratios are calculated from beta, they are not typically reported separately from the odds ratio. We have also explained that following recent guidelines from the American Statistical Association, we focus on effect size rather than relying on statistical significance. We have added text explaining that trust is statistically significantly associated with each of the outcomes, but we prefer that the reader rely on the effect size (odds ratio and confidence intervals presented in the tables).

We hope that this meets with your approval. Thank you.

Reviewer 2 Report

The topic of trust within an organizational context is highly significant. Much has been made about the association between the workplace climate and mental health outcomes; however, not a great deal of research has been devoted to examining health outcomes. This research lines clearly establishes the continuing need to examine this relationship. 

Author Response

Thank you for your supportive comments. We hope that our article contributes significantly to the literature.